# Refolding Increases the Chaperone-like Activity of α_H_-Crystallin and Reduces Its Hydrodynamic Diameter to That of α-Crystallin

**DOI:** 10.3390/ijms241713473

**Published:** 2023-08-30

**Authors:** Konstantin O. Muranov, Nicolay B. Poliansky, Vera A. Borzova, Sergey Y. Kleimenov

**Affiliations:** 1Emanuel Institute of Biochemical Physics of Russian Academy of Sciences, Moscow 119334, Russia; npoliansky@mail.ru; 2Federal Research Centre “Fundamentals of Biotechnology” of Russian Academy of Sciences, Bach Institute of Biochemistry, Moscow 119334, Russia; vera.a.borzova@gmail.com; 3Koltzov Institute of Developmental Biology of Russian Academy of Sciences, Moscow 119334, Russia; s.yu.kleymenov@gmail.com

**Keywords:** alpha-crystallin, alphaH-crystallin, high molecular weight aggregate, chaperone, chaperone-like activity, refolding, urea

## Abstract

α_H_-Crystallin, a high molecular weight form of α-crystallin, is one of the major proteins in the lens nucleus. This high molecular weight aggregate (HMWA) plays an important role in the pathogenesis of cataracts. We have shown that the chaperone-like activity of HMWA is 40% of that of α-crystallin from the lens cortex. Refolding with urea significantly increased—up to 260%—the chaperone-like activity of α-crystallin and slightly reduced its hydrodynamic diameter (*D*_h_). HMWA refolding resulted in an increase in chaperone-like activity up to 120% and a significant reduction of *D*_h_ of protein particles compared with that of α-crystallin. It was shown that the chaperone-like activity of HMWA, α-crystallin, and refolded α-crystallin but not refolded HMWA was strongly correlated with the denaturation enthalpy measured with differential scanning calorimetry (DSC). The DSC data demonstrated a significant increase in the native protein portion of refolded α-crystallin in comparison with authentic α-crystallin; however, the denaturation enthalpy of refolded HMWA was significantly decreased in comparison with authentic HMWA. The authors suggested that the increase in the chaperone-like activity of both α-crystallin and HMWA could be the result of the correction of misfolded proteins during renaturation and the rearrangement of protein supramolecular structures.

## 1. Introduction

Lens opacity, or cataracts, is the leading cause of low vision and blindness in the world [1,2]. The key role of short-range interactions of protein in the formation of lens opacity was predicted in 1971 by George Benedek and experimentally confirmed only 50 years later [3]. It was shown that lens opacity is caused by fluctuations in the protein concentration in the cytoplasm of fiber cells and the formation of so-called multilamellar bodies [4,5].

More than 90% of soluble lens proteins are α-, β-, and γ-crystallins. The protein composition of the lens is heterogeneous: fractions of high-molecular forms of α- and β-crystallin and the monomolecular form of γ-crystallin predominate in the nucleus, while α- and β_L_-crystallin are the main proteins in the cortex [6]. There is little or no turnover of lens proteins, which leads to the accumulation of post-translational modifications that destabilize proteins with age [7]. 

α-Crystallin as a chaperone-like protein inhibits the aggregation of damaged proteins and thus maintains the transparency of the lens [8,9]. The structure and function of this protein have been discussed in detail in recent reviews [10,11,12]. This polydisperse oligomeric molecule, formed by monomers of αA-crystallin and αB-crystallin, has a mass of 160 to 1000 kDa and is capable of blocking the aggregation of destabilized proteins. This type of α-crystallin is located predominantly in the lens cortex, whereas its high-molecular form, α_H_-crystallin, predominates in the nuclear part of the lens. The mass of α_H_-crystallin is several million daltons. Given that the main light flux passes through the nuclear region of the lens, the study of the role of α_H_-crystallin in the development of opacity is extremely important.

Electron microscopic studies have shown that α_H_-crystallins appear as very large, nonspherical, randomly shaped aggregates of protein particles with a size slightly smaller than α-crystallin of the lens cortex [13,14]. In other words, α_H_-crystallin is a high molecular weight aggregate (HMWA). Hereafter, we will use this term to emphasize that α_H_-crystallin is a protein particle stochastically assembled from α-crystallin oligomers. It should be clarified that in this case, α-crystallin is a modified protein. Not only do its monomers have post-translational modifications, but the oligomer itself can include damaged lens proteins, for example, γ-crystallin. The size of HMWA increases with age and its relative amount also increases [15,16,17]. Post-translational modifications of αA- and αB-crystallin are presumed to lead to the formation of HMWA. However, direct damage to α-crystallin by ultraviolet light did not lead to the formation of HMWA [18]. The presence of other proteins is probably also required for the formation of HMWA [19]. Studies of the circular dichroism, absorption, and fluorescence spectra of HMWA showed that HMWA had a different tertiary structure from that of α-crystallin, with HMWA being more unfolded [20,21,22,23]. 

Using a model of thermal aggregation of proteins, it was shown that the chaperone-like activity of HMWA was reduced [24,25]. On the other hand, in the model of dithiothreitol-induced insulin aggregation, it was shown that the activity of HMWA did not differ from that of α-crystallin from the cortex [26]. However, the methods used to measure the chaperone-like activity were not quantitative. 

There are data indicating that the chaperone-like activity of HMWA depends on changes in the tertiary and quaternary structure but not on the disruption of the secondary structure of the protein [24]. At the same time, it was shown that α-crystallin point mutations associated with the occurrence of congenital cataracts and causing a decrease in chaperone-like activity cause significant changes in the secondary, tertiary, and quaternary structure of the protein [27,28]. Therefore, it can be expected that the chaperone-like activity of HMWA also depends on the secondary structure of the protein. 

It was shown that refolding of α-crystallin under the action of urea changes the tertiary and quaternary structure but not the secondary structure of the protein [29,30]. Therefore, protein refolding in the presence of urea was used to study the role of tertiary and quaternary structures in the chaperone-like function of HMWA. 

In recent years, methods have been developed to quantify the chaperone-like activity of both proteins and chemical chaperones based on kinetic analysis of aggregation [31,32,33,34,35]. 

The aim of this work was to quantify the chaperone-like activity of native and renatured HMWA versus native and renatured α-crystallin from the lens cortex and compare this parameter with a portion of native protein in the HMWA and α-crystallin preparations. UV-damaged β_L_-crystallin was used as the target protein for the determination of the chaperone-like activity of HMWA and α-crystallin. Using UV-damaged β_L_-crystallin made it possible to carry out measurements under conditions close to physiological ones [36,37].

It was shown the chaperone-like activity of HMWA was 40% of the activity of α-crystallin. Refolding did not affect the structure of α-crystallin but significantly increased the chaperone-like activity, up to 260%. HMWA refolding resulted in an increase in the chaperone-like activity up to 120% and a significant reduction in the hydrodynamic diameter of protein particles compared with that of α-crystallin. Surprisingly, the low concentrations of the renatured HMWA accelerated aggregation of UV-damaged β_L_-crystallin, but aggregation was inhibited by an increase in the concentration of renatured HMWA. The chaperone-like activity of the tested samples strongly correlated with the portion of the native protein of these samples, excluding refolded HMWA. We suggest that the increase in the chaperone-like activity of both α-crystallin and HMWA could be a result of the correction of misfolded protein molecules during renaturation and the rearrangement of protein supramolecular structure. The structural rearrangement is, apparently, of greatest significance in the case of refolded HMWA.

## 2. Results and Discussion

### 2.1. Refolding of α-Crystallin and HMWA

#### 2.1.1. Study of a Size of α-Crystallin and HMWA Particles under Urea-Induced Denaturation and Renaturation

HMWA is a conglomerate of particles close to α-crystallin in size [38]. Since age-related changes did not significantly affect HMWA chaperone-like activity, Carver et al. suggested that post-translational modifications did not affect HMWA chaperone-like function [24]. It is possible that the formation of the α-crystallin conglomerate is a result of accumulation of post-translational modifications. Therefore, we carried out the procedure of denaturation-renaturation of the studied proteins in the presence of urea. 

The dependencies of α-crystallin and HMWA particle hydrodynamic diameter (*D*_h_) measured by dynamic light scattering (DLS) on the urea concentration are shown in Figure 1. The black circles show the particle *D*_h_ during protein denaturation, and the open circles show the particle *D*_h_ during renaturation, i.e., when the urea concentration decreases. The *D*_h_ of α-crystallin is 19.5 nm, which corresponds to the molecular weight of about 750 kDa, and the *D*_h_ of HMWA is 43 nm, which corresponds to 4500 kDa. With an increase in the concentration of urea, the particle *D*_h_ decreased, with a sharp transition in the region of 3–4 M. Further, the particle *D*_h_ changed insignificantly. The *D*_h_ of α-crystallin at the maximum concentration of urea was 7.0 ± 0.8 nm, and the particle *D*_h_ of HMWA at the maximum concentration of urea was 8.0 ± 1.2 nm. This is consistent with the literature data [30,39]. The molecular mass of such particles, calculated for a linear polymer, is 20–22 kDa. This indicates that in the presence of urea, the α-crystallin and HMWA oligomers dissociate into monomers. 

The *D*_h_ of renatured α-crystallin was slightly lower than the *D*_h_ of the native oligomer and equal to 17 nm, which coincides with the literature data [29,40]. Refolding caused a twofold decrease in the *D*_h_ of HMWA. The *D*_h_ of the formed particles was equal to the *D*_h_ of α-crystallin (19.5 nm and 23 nm, respectively). Hereinafter, we will refer to such a preparation as HMWAr.

Figure 2A shows the molecular and polypeptide composition of HMWA and HMWAr. 

HMWA eluted as a single peak with a mass of 2000 kDa when separated on Sephacryl S300 HR. The difference in the mass of HMWA measured with gel filtration and DLS was the result of the gel used. The upper limit of Sephacryl S300 resolution was 1500–2000 kDa. HMWAr was separated into three fractions with masses of 900, 1700, and 2300 kDa. The broad peak with a maximum of 900 kDa was the main one. The half-width of this peak indicates that the fraction was polydisperse and probably consisted of several components with masses between 300 and 1200 kDa.

The image analysis of the SDS PAGE (Appendix A) shows the polypeptide composition of α-crystallin, which does not differ from standard preparations [37]. It consists of αA-crystallin (30%), αB-crystallin (60%), and truncated α-crystallin (10%). Figure 2B demonstrates that the polypeptide composition of different HMWA fractions is almost the same. The polypeptide composition of the fractions is shown in Appendix A. Compared with α-crystallin, HMWA polypeptides can be divided into three groups. Group #1: αA-crystallin (20 kDa), αB-crystallin (19.8 kDa), and a mixture of their truncated forms (19.4 kDa); polypeptides were identified by mass. These substances make up about 85% of the total amount of polypeptides. Group #2: polypeptides with masses greater than 20 kDa. They range from 8% to 12%. Group #3: polypeptides with masses of 6 to 12 kDa. Their amount was approximately 7%. The polypeptide composition of α-crystallin shows no difference from standard preparations [36]. It consists of αA-crystallin (30%), αB-crystallin (60%), and truncated α-crystallin, which is a mixture of αA-crystallin and αB-crystallin fragments. The polypeptide composition of the fractions (Appendix A) is shown in Appendix A (Appendix A). The presence of proteins with a mass that is not a multiple of 20 kDa indicates that not only modified α-crystallin but also other proteins are included in HMWA. Identification of group 2 and 3 polypeptides was not carried out as it was not necessary for achieving the main objective of the study.

#### 2.1.2. Study of Thermostability of Native and Refolded α-Crystallin and HMWA by DSC 

We used DSC to investigate the stability of the protein structure of α-crystallin and HMWA upon refolding. This method, while not reflecting changes in the oligomeric assembly of protein complexes, provides valuable information about the lower levels of the protein structure.

The DSC profiles of both native and refolded α-crystallin and HMWA are shown in Figure 3. The denaturation enthalpies and temperatures (T_max_) are given in Table 1. It was assumed that the heat of denaturation, expressed by the area under the DSC profile, was proportional to the amount of the native protein [41]. 

The T_max_ value did not differ between native and renatured α-crystallin. This strongly suggests that the protein structure was not significantly altered during refolding, which is supported by the literature data [29]. At the same time, refolding increased the denaturation enthalpy of α-crystallin by 40%, which indicates an increase in the amount of native protein in the sample (Figure 3A, Table 1). The denaturation enthalpy of HMWA was reduced relative to native α-crystallin by 31%. The shift of the maximum HMWA denaturation peak toward lower temperatures by 0.8 °C compared with α-crystallin may be due to the superposition of the native α-crystallin peak of denaturation and some other peak, indicating the presence of an additional structure with a slightly lower thermostability. A decrease in the enthalpy of denaturation indicated the presence of denatured or partially denatured protein molecules in HMWA [41].

The HMWA refolding is of particular interest. In our experiment, proteins from the lens of steers were used in order to reduce age-related accumulation in HMWA of proteins destabilized by post-translational modifications [24]. However, the denaturation enthalpy of HMWAr was significantly (by 35%) lower than that of native α-crystallin. This indicated that the structure of the protein molecules comprising HMWAr was significantly changed, and they may have lost the ability to refold properly. The asymmetry of the DSC profile of HMWAr and the decrease in T_max_ by 1.5 °C also warrants attention. 

It should be noted that changes in the thermal stability of α-crystallin and β_L_-crystallin are significantly different from changes in the thermal stability of other proteins under UV-induced oxidative stress, which is the main cause of damage for eye lens proteins. UV irradiation did not cause a change in T_max_ of α-crystallin and β_L_-crystallin [37,41], while T_max_ of lactate dehydrogenase (LDG), glyceraldehyde-3-phosphate dehydrogenase (GAPHD), and glycogen phosphorylase *b* (Ph*b*) significantly decreased depending on the UV dose. A decrease in T_max_ resulted from the formation of more thermolabile forms of these proteins [35,42,43,44]. By analogy with the above results for LDG, GAPHD, and Ph*b*, we can assume that as a result of denaturation-renaturation, part of the protein in the HMWA preparation passed into a more thermolabile form. 

Thus, as a result of refolding, HMWA underwent significant changes, which were expressed as a decrease in both the amount of native protein and the protein particle *D*_h_.

### 2.2. Investigation of the Chaperone-like Activity of HMWA in Comparison with α-Crystallin

#### 2.2.1. Kinetics of UV-Damaged β_L_-Crystallin Aggregation

The irradiation conditions for β_L_-crystallin (irradiation temperature, power, and dose of ultraviolet light) were selected in such a way that denatured UV-damaged βL-crystallin molecules prone to aggregation accumulated in the solution (Appendix A). The aggregation of such damaged molecules was initiated with an increase in temperature to 37 °C [33] (Appendix A). Note that UV-damaged β_L_-crystallin used in further experiments was a 1:1 mixture of denatured and native proteins, although native β_L_-crystallin does not aggregate at this temperature. The general view of the aggregation kinetic curves of UV-damaged β_L_-crystallin at different protein concentrations is shown in Figure 4A. It should be noted the SPECTROstar Nano optical density range was 0 to 4 OD and the photometric resolution was 0.001 OD. We note two features of the curves associated with the use of a plate spectrophotometer for measuring aggregation. (1) The extended lag period was due to the fact that a cold plate (4 °C) was placed in the spectrophotometer. Therefore, it took time to heat the sample to 37 °C. (2) The strong noise at the terminal stage of aggregation at a protein concentration above 1.25 mg/mL was caused by the following circumstances: (i) the precipitation of protein aggregates; (ii) the vertical direction of the measuring beam of light, the fluctuations of which were enhanced upon mixing the sample during the movement of the microplate carrier. It is necessary to note that in further analysis we used the initial parts of the kinetic curves and β_L_-crystallin concentration of 1.5 mg/mL.

UV-induced aggregation is a multi-step process that consists of the following steps: (1) UV-induced damage of protein; (2) protein denaturation; (3) nucleation stage; and (4) aggregate growth stages [37]. Protein aggregates formed in step 4 are the product of this reaction. The use of kinetic analysis requires that the measured parameter be linearly related to the amount of the reaction product. 

Figure 4B shows that the apparent optical density of the incubation mixture depends linearly on the concentration of the aggregated protein. This means that this parameter can be used for kinetic analysis of aggregation. It should be noted that the apparent optical density of a solution is not always a measure of the amount of the reaction product (amount of aggregated protein). For example, in the aggregation of UV-damaged Ph*b*, the amount of aggregated protein is linearly related to light scattering intensity but not to apparent optical density [45].

Analysis of the aggregation kinetics shows that after completion of the nucleation stage (i.e., at the stage of aggregate growth), the kinetics follows an exponential law (Figure 4C), indicating the first order of protein aggregation. Thus, the order of aggregation by protein concentration is calculated from the individual kinetic curve, n_t_ = 1 [46]. Figure 5 shows that the dependences of the main parameters *A*_lim_, *k*_I_, and *t*_0_ on the protein concentration are reliably described by a linear dependence. The presence of a lag period on the kinetic curves, the duration of which decreases with increasing protein concentration (Figure 5C), indicates a mechanism of nucleation-dependent polymerization. The linear dependence of *A*_lim_ on protein concentration confirms that the apparent optical density is directly proportional to the concentration of the aggregated protein for all the protein concentrations studied. However, it appears that the rate constant *k*_I_ increases linearly with increasing protein concentration [P]_0_. This indicates that k_I_ is the rate constant of a pseudo-first-order reaction. 

The product *k*_I_·*A*_lim_ can be considered as a measure of the initial rate of aggregation. It was observed, for example, in thermal aggregation of tobacco mosaic virus coat protein [47], firefly luciferase [48], thermal aggregation of rabbit muscle creatine kinase [49], and guanidine hydrochloride-induced aggregation of Ph*b* from rabbit skeletal muscles [50]. For UV-damaged β_L_-crystallin, the initial aggregation rate *k*_I_ ·*A*_lim_ is proportional to the protein concentration squared (Figure 6A). We designate the order of aggregation with respect to protein calculated from the dependence of the initial rate of aggregation on the initial protein concentration as n*_C_* [46]. These data indicate that in the considered cases the n*_C_* value was equal to 2. 

To explain the situation, when n*_t_* = 1 and n*_C_* = 2, we consider the expression for the rate of the growth of aggregates [51,52]:υ = *k_II_*·[nucleus][D],(1)
where k*_II_*—the rate constant of the second order, [nucleus]—concentration of nuclei, i.e., protein microaggregates, [D]—concentration of denatured protein. Both the concentrations of nuclei and denatured protein are proportional to the initial protein concentration [P]_0_ through the respective rate constants of nucleation and denaturation. Therefore, the initial rate for the stage of aggregate growth was proportional to protein concentration squared (n*_C_* = 2). On the other side, we assume that as we approach the end of the nucleation stage, the concentration of nuclei reaches a constant value. The further growth of aggregates continues with the formation of denatured protein molecules and their attachment to the nuclei. In this case, the product k*_II_*· [nucleus] is transformed into the pseudo-first-order rate constant k*_I_*. This constant depends on the protein concentration (as shown in Figure 5B) due to the proportionality of the final nuclei concentration to [P]_0_, whereas the initial rate of aggregation depends linearly on the protein concentration due to the proportionality of [D] to [P]_0._ The concentration of monomer D in time decreases in accordance with the exponential law (n*_t_* = 1). Thus, for the aggregation kinetics of UV-damaged β_L_-crystallin, there are n_t_ = 1 and n_C_ = 2. 

In the case of UV-damaged β_L_-crystallin, a single-hit model of denaturation under UV light was performed [37]. This was indicated by the fact that the shape of the DSC profile remained unchanged with increasing radiation dose. The case is quite different for UV-irradiated Ph*b* and GAPHD: n_t_ = 1 and n_C_ = 1 [45,53]. The single-hit model of UV-induced denaturation was not performed for these proteins. This was indicated by the shift of the DSC profile T_max_ toward lower temperatures with increasing irradiation dose. UV irradiation leads to the formation of latent damaged states. The structural rearrangement of these latent damaged states occurs upon heating, i.e., the rate of this monomolecular stage controls the rate of aggregation. Compared with these cases, the test system based on the aggregation of UV-damaged β_L_-crystallin makes it possible to study the direct effect of various agents on protein aggregation.

#### 2.2.2. Chaperone-like Activity of Native and Refolded α-Crystallin and HMWA

The initial adsorption capacity (*AC*_0_) of the chaperone was used as a measure of the chaperone-like activity [41]. Figure 7 shows the dependence of the relative aggregation rate on the molar ratio of the chaperone to the target protein. The dependences of υ/υ_0_ on the chaperone/target protein ratio were similar for native and refolded α-crystallin and native HMWA. Namely, at the initial section of the curve, there was a rapid decrease in the aggregation rate; then, the decrease slowed down, which is consistent with previously published results [41,54,55]. The dependence for HMWAr was significantly different. The addition of HMWAr in small amounts to UV-damaged β_L_-crystallin caused a significant acceleration of aggregation (Figure 7B). However, with an increase in the amount of HMWAr, inhibition of aggregation was observed. Therefore, the points marked with asterisks were excluded from the analysis to calculate *AC*_0_. 

Fitting the initial section of the dependence with a linear equation allowed us to calculate *AC*_0_ as a segment cutoff on the *x*-axis. Table 2 presents the obtained values. The *AC*_0_ value indicated that one α-crystallin monomer can bind 5 UV-damaged β_L_-crystallin molecules, while one HMWA monomer can bind only 2 molecules of the target protein. Thus, the chaperone-like activity of HMWA was 40% of that of α-crystallin from the lens cortex.

Refolding reduced the *D_h_* of the α-crystallin oligomer from 19.5 to 17 nm. This indicates an increase in the amount of oligomers in the sample since they were formed from the same amount of monomers. The decrease in α-crystallin *D*_h_ can be explained by an increase in the compactness of the oligomer. However, an increase in the chaperone-like activity of α-crystallin, on the contrary, requires loosening of the structure of protein chains, i.e., increasing the availability of hydrophobic sites for target protein binding [56]. 

The simplest geometric calculation shows that as the hydrodynamic diameter of the oligomer decreases by 1.5 nm, the number of oligomers increases by a factor of 1.5. This may mean that the chaperone-like activity of such a protein also increases. Assuming that, *AC*_0_ also should increase by a factor of 1.5. It has been shown that refolding also affects the ability to bind hydrophobic substrates. In particular, it was found that refolding increases the number of bisANS binding sites by a factor of 1.2 [29]. We believe that the total effect of these factors on the increase in the chaperone-like activity is 1.5 × 1.2 = 1.8. However, the increase in *AC*_0_ was 2.6-fold (Table 2). This indicates the presence of another factor that enhances the chaperone-like activity of α-crystallin.

Figure 8 shows the dependence of *AC*_0_ on the denaturation enthalpy for the studied proteins. The dependence fits into an exponent with a high correlation, only the data for HMWAr was omitted. The obtained results correspond well with the data for UV-irradiated α-crystallin, the chaperone-like activity of which decreased exponentially with increasing enthalpy of denaturation [41]. Thus, we can assume that the chaperone-like activity of both α-crystallin and HMWA is determined by the portion of the structured protein in the molecule. The decrease of *AC*_0_ value in the case of HMWAr indicates that its structure was changed during the renaturation process. This disturbance may be the result of the formation of some other, more energetically favorable, protein structures, for example, protein aggregates. 

The protein oligomers isolated from the lens may have various structure defects. These may be secondary structure defects resulting from folding errors during protein synthesis. Such defects can also be formed due to the influence of various damaging factors that can impair the tertiary structure of the protein [57,58]. These may also be irregularities in the quaternary structure since the exchange of subunits is an integral characteristic of α-crystallin. We suppose that all these defects can be repaired during the refolding process. 

The evidence is an increase in the total enthalpy of thermodenaturation by 40% (Table 1), which means an increase in the proportion of native protein in the renatured α-crystallin. This assumption also confirms the restoration of the native structure of proteins, in particular enzymes, during refolding in the presence of α-crystallin [59,60]. In other words, α-crystallin is a chaperone for itself.

At ratios of HMWAr/UBβ_L_-crystallin equal to 1/13 and 1/7 (mol/mol), an acceleration of aggregation was found. This may mean that the HMWAr sample contains oligomers capable of forming water-insoluble complexes with the denatured protein. Some monomers that make up HMWA can be damaged due to post-translational modifications. This can prevent the correct folding of protein chains during renaturation. It is likely that some monomers with a large amount of damage adopt a conformation in which hydrophobic fragments of the polypeptide chain appear on the surface of the globule. Such proteins are prone to aggregation, and in their presence, the rate of aggregation increases. Since the chaperone-like activity of HMWA is reduced, its protective properties appear only at ratios of HMWAr/UV-damaged β_L_-crystallin equal to or greater than ¼ (Figure 7B). Presumably, these proteins with folding defects are the ones that form the high molecular mass subfractions of 1700 and 2300 kDa in HMWAr (Figure 2A).

The results of this study allow us to suggest that the refolding of α-crystallin and HMWA in the presence of urea can correct at least some of the accumulated structural damage in the protein molecules.

## 3. Materials and Methods

### 3.1. Materials

Lenses of 6-month-old steers were sourced from a local slaughterhouse. Sephacryl S300 HR, Sephacryl S200 superfine, molecular weight protein standards, i.e., thyroglobulin, ferritin, aldolase (158 kDa), BSA dimer (134 kDa), BSA (67 kDa), egg albumin (45 kDa), cytochrome C (12.5 kDa), and EDTA were produced by Sigma-Aldrich. The solution of urea, Sigma-Aldrich (St. Louis, MO, USA) was purified from its decay product with AG 501-X8 mixed bed resin. BioRad (Hercules, CA, USA). Reagents for SDS PAGE were produced by BioRad. NaCl, NaN_3_ Na_2_HPO_4_, NaH_2_PO_4_·H_2_O were provided by PanEco (Moscow, Russia). Millipore PTTK disk membrane, NMWL 30 000, Merk (Merck & Co., Inc., Rahway, NJ, USA), was also used.

### 3.2. Isolation and Purification of Crystallin

Purification of α-crystallin, α_H_-crystallin, and β_L_-crystallin from the lenses of 6-month-old steers was performed according to the procedure described earlier [61]. The decapsulated lenses were slowly stirred in PBS (40 mM sodium phosphate, pH 7.0, containing 100 mM NaCl, 1 mM EDTA, and 3 mM NaN_3_) at 0 °C using a magnetic stirrer to separate the lens nucleus and cortex. The lens nucleus and cortex were homogenized at 0 °C in PBS. The homogenates of cortex and nucleus were centrifuged at 27,000× *g* for 1 h at 4 °C, and the supernatant containing soluble crystallins was fractionated by gel filtration using Sephacryl S300 HR (Sigma-Aldrich) column. The peaks corresponding to α_H_-crystallin, α-crystallin, and β_L_-crystallin were detected by measuring absorbance at 280 nm and collected.

Further purification of α_H_-crystallin and α-crystallin was achieved by rechromatography of crystallin-containing fractions using the Sephacryl S300HR (Sigma-Aldrich) column. Further purification of β_L_-crystallin was achieved by repeated chromatographic separation of the crystallin-containing fractions using the Sephacryl S200 superfine (Sigma-Aldrich) column. Finally, the peak corresponding to the desired protein was collected and concentrated by ultrafiltration (Millipore PTTK disk membrane (Sigma-Aldrich), NMWL 30 000). The protein concentration was determined spectrophotometrically at 280 nm using the absorption coefficient A_1%_ 1 cm of 0.85 and 2.3 for α- and β_L_-crystallin, respectively [62], and by a micro biuret method [63]. Approximately 50 mg of α_H_-crystallin, α-crystallin, and β_L_-crystallin with an average concentration of 20 mg/mL were obtained per isolation cycle. The protein solutions were stored at 7 °C under an argon atmosphere.

### 3.3. Denaturation and Renaturation of α-Crystallin and HMWA

To study the effect of denaturation and renaturation on the particle size of α-crystallin and HMWA, the protein samples with different concentrations of urea were used. The denatured protein samples at urea concentrations from 1 to 7.6 M were prepared by mixing a solution of native protein in PBS (10 mg/mL) with a urea solution (8 M) so that the final protein concentration was 1 mg/mL, and urea concentration was varied stepwise. The samples were incubated for 4 h at room temperature and then stored in a refrigerator at 7 °C under argon. Renatured samples at various concentrations of urea were obtained as follows. Aliquots of denatured protein (8 mg/mL) in 8 M urea were diluted with PBS to different final concentrations of urea (from 1 to 7.6 M) and constant protein concentration (1 mg/mL). The mixtures were incubated at room temperature for 4 h, then the samples were placed in a refrigerator and stored at 7 °C under argon. 

Renatured α-crystallin and HMWA for the DSC experiments, the study of molecular and polypeptide composition, and the measurement of chaperone-like activity were carried out as follows. The solution of the corresponding native protein in PBS was mixed with 8 M urea (pH = 7.0) and incubated for 4 h. The final concentration of protein was 8 mg/mL, and the final concentration of urea was 6 M. A sample of denatured protein was diluted 1/10 (*v*/*v*) with PBS and concentrated using Amicon^®^ Ultra 30K devices on a Beckman J-6 centrifuge. The washing procedure was repeated three times. The final concentration of the renatured protein was 10 mg/mL.

### 3.4. Dynamic Light Scattering

The *D*_h_ of protein particles was measured using the dynamic light scattering on a ZetaSizer S nano instrument (Malvern instruments) in automatic mode. The hydrodynamic diameter (*D*_h_) and molecular mass of the protein particle were calculated using the built-in software (Zetasizer Software 7.10) taking into account the change in viscosity and refractive index of the solvent at different urea concentrations [64].

### 3.5. Size Exclusion Chromatography

The samples of native and refolded α_H_-crystallin were centrifuged at 14,500× *g* for 30 min using a MiniSpin+ Eppendorf centrifuge. The supernatant was loaded onto the column (Sephacryl S300HR, 2.5 × 90 cm) and separated into fractions at a flow rate of 1.6 mL/min. The column was pre-calibrated with the following proteins from (Sigma-Aldrich): thyroglobulin (660 kDa), ferritin (440 kDa), aldolase (158 kDa), BSA dimer (134 kDa), BSA (67 kDa), egg albumin (45 kDa), and cytochrome c (12.5 kDa). The standard error for protein mass determination was 4%.

### 3.6. Sodium Dodecyl Sulphate Polyacrylamide Gel Electrophoresis (SDS-PAGE)

The polypeptide composition of α-crystallin and α_H_-crystallin was analyzed with electrophoresis in 15% PAGE in the presence of SDS and DTT (50 mM) [65]. Sigma-Aldrich proteins with known molecular masses were used as standards. The gels were stained with Coomassie R-250 and scanned with photoscanner Epson Perfection 4180. The images were analyzed with the ImageJ 1.52v program.

### 3.7. Calorimetric Studies 

Thermal denaturation of native or renatured α-crystallin and α_H_-crystallin was studied with DSC using a MicroCal VP-DSC microcalorimeter (Malvern Instruments, Northampton, MA, USA) in 40 mM sodium phosphate, 100 mM NaCl, 1 mM EDTA, 3 mM NaN_3_, pH 7.0. The protein solution was heated from 5 to 90 °C at a constant rate of 1 °C/min and a constant pressure of 2.2 atm. The irreversibility of the thermal transition of proteins was proved by checking the reproducibility of the calorimetric trace. To do this, the sample of the protein was scanned twice in a row without removing it from the calorimeter. Calorimetric traces of irreversibly denaturing proteins were corrected for instrumental background by subtracting buffer vs. buffer scans. The temperature dependence of the excess heat capacity was further analyzed. 

### 3.8. UV Irradiation of β_L_-Crystallin Preparation

UV irradiation of β_L_-crystallin (2 mg/mL) was carried out in a 1 cm path quartz cell at 8 ± 2 °C. The volume of the sample was 2 mL. The sample was slowly (30 rpm) stirred using a stainless steel mixer. UV irradiation was applied using UV LED (232S-S6060 UVC Quad SMD LED Modul, Laser Components GmbH, Germany (Olching, Germany). Incident light spectra and irradiation device drawing are shown in Appendix A. The UV output incident was measured with an optical spectrometer AvaSpec-2048. The summary incident power was calculated as an integral of 260–305 nm intervals and was equal to 70.0 ± 0.1 μW/cm^2^. The irradiation dose was 140 ± 5 mJ/cm^2^. 

### 3.9. Optical Density of Incubation Medium and the Amount of Aggregated Protein

Aggregation of UV-damaged β_L_-crystallin was carried out at 37 °C using a SPECTROstar^®^ Nano plate spectrophotometer at 360 nm. Samples of UV-damaged β_L_-crystallin solution from 0.15 to 1.0 mg/mL were aggregated upon reaching the maximum value of the apparent optical density. The sample was quickly cooled in ice and immediately centrifuged for 15 min at 14,500× *g* and 4 °C. The precipitate was washed twice with cold buffer and dissolved in 6% NaOH. The amount of protein in sediment was measured using the micro biuret method [63]. The dependence of optical density on the concentration of the aggregated protein was investigated. The measurements were repeated three times.

### 3.10. Study of the Aggregation Kinetics of UV-Damaged β_L_-Crystallin

The aggregation kinetics was studied using a SPECTROstar^®^ Nano plate spectrophotometer at 360 nm [66]. PBS, chaperone solution, and UV-damaged β*_L_*-crystallin solution were added to a plate well (Greiner Bio-One UV-STAR^®^ PLATE) at 4 °C. The final concentration of the UV-damaged β*_L_*-crystallin sample varied from 0.1 to 4 mg/mL, and the sample volume was 100 µL. The aggregation kinetics was recorded at a frequency of 0.1 Hz. The kinetics curves were analyzed using the method developed by B.I. Kurganov and described in detail in the papers ([41,67]. Briefly, the section of the aggregation curve after the inflection was approximated by a first-order reaction equation:(2)$$A=A_{\lim}\cdot (1-e^{-k_{I}\cdot (t-t_{0})}),$$
where *t*_0_ is a length on the horizontal line at *A* = 0 cutoff by the theoretical curve calculated with this equation; *A* is the apparent optical density; *A*_lim_ is the limiting value of *A* at *t* → ∞; and *k_I_* is the rate constant of the first-order reaction.

Studying the dependences of *A*_lim_, *k*_I_, [*A*_lim_*·k*_I_], on the concentration of UV-damaged β_L_-crystallin, we calculated the order of aggregation by protein. The order of protein aggregation calculated from individual aggregation curves was denoted as n_t_.

Using the dependence of the initial aggregation rate on protein concentration, we calculated the order of aggregation for the protein, which was denoted as n_C_.

The initial aggregation rate was calculated as:*υ*_0_ = *A_lim_·k*_I_, (3)
where *υ*_0_ is the initial rate of the visible stage of the process or by fitting the section of the aggregation curve after the inflection by the equation:*A = υ*_0_ (*t* − *t*_0_) − *B(t* − *t*_0_)^2^,(4)
where 

*A*—apparent optical density of the solution*t*—time*t*_0_—the duration of the nucleation stage *υ*_0_—the initial rate of the visible stage of aggregation *B*—constant

### 3.11. Measurement of the Chaperone-like Activity of α-Crystallin and HMWA

The adsorption capacity of a protein chaperone with respect to the target protein can be considered as a measure of the anti-aggregation activity of the chaperone [54]. To calculate the adsorption capacity of the chaperone, the dependence of the value (*υ*/*υ*_0_)^1/n^ on the ratio of the molar concentrations of the chaperone and the target protein (*x* = [chaperone]/[target protein]) has been plotted (*υ*/*υ*_0_ is the relative initial rate of aggregation of the target protein; *υ*_0_ is the initial rate of aggregation in the absence of the chaperone). The initial adsorption capacity of the chaperone (*AC*_0_) has been calculated from the initial part of the dependence of (*υ*/*υ*_0_)^1/n^ on *x*:(5)$${\left(\frac{\upsilon }{\upsilon_{0}}\right)}^{1/n}=1-AC_{0}\cdot x$$

The length cutoff on the abscissa axis by the linear part of the dependence of (*υ*/*υ*_0_)^1/*n*^ on *x* corresponds to the reciprocal value of *AC*_0_. The *x*-ratio was calculated for monomers of α-crystallin and HMWA.

### 3.12. Statistic Analysis

The descriptive statistics and nonparametric Mann-Whitney U test were used for data analysis (StatSoft, Attestat 13.11 © I.P. Gaydyshev, Tomsk, Russia). Package Origin Pro 2015 was used to fit kinetic curves. The degree of agreement of experimental data and calculated values was characterized by the coefficient of determination R^2^ (© Origin Lab Corp, Northampton, MA 01060, USA).

## 4. Conclusions

The structure and function of chaperone-like protein α-crystallin maintains the transparency of the lens. α_H_-Crystallin, i.e., the high molecular weight form of α-crystallin, is mostly located in the lens nucleus whereas α-crystallin prevails in the lens cortex. Given that the main light flux passes through the nuclear region of the lens, the study of the role of α_H_-crystallin in the development of lens opacity (cataract) is extremely important. 

Using a model of aggregation of UV damaged β_L_-crystallin and the quantitative method of kinetic analysis shows that the chaperone-like activity of α_H_-crystallin was 40% of the activity of α-crystallin from the lens cortex. Urea-induced refolding strongly increased the chaperone-like activity of α-crystallin and slightly increased that of α_H_-crystallin. Refolded α_H_-crystallin at low chaperone/target protein ratios caused an acceleration of aggregation, which was associated with the formation of protein particles prone to aggregation during renaturation. Increasing the amount of renatured HMWA above a certain value inhibited aggregation.

Urea-induced refolding slightly reduced the *D*_h_ of the α-crystallin but caused a significant rearrangement of the quaternary structure of α_H_-crystallin. Refolding converted α_H_-crystallin into a protein particle, the *D*_h_ of which was approximately equal to the *D*_h_ of authentic α-crystallin. 

It was revealed that the chaperone-like activity was strongly correlated with a portion of native protein in the α-crystallin, refolded α-crystallin, and α_H_-crystallin. Refolded α_H_-crystallin was an outlier in this dependency. It is hypothesized that renaturation with urea repairs misfolded proteins, which are formed both during protein synthesis and during the functioning of the protein in the cell.

The results obtained allow us to suggest that the refolding of α-crystallin and α_H_-crystallin in the presence of urea can correct at least some of the accumulated structural damage in the protein molecules. However, if a level of post-translational modifications of protein exceeds a certain threshold, then part of the protein not only loses its chaperone-like activity but generates forms prone to aggregation. The other part of α_H_-crystallin partially restores the native structure and demonstrates an increased chaperone-like activity. Our results may contribute to fundamental research in understanding the role α- and α_H_-crystallin in lens physiology.

## Figures and Tables

**Figure 1 ijms-24-13473-f001:**
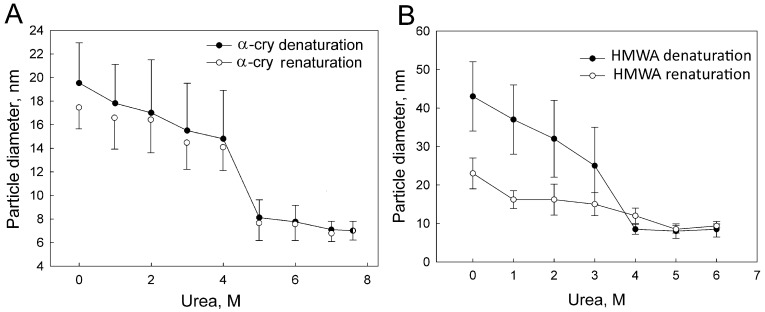
The dependencies of protein particles *D*_h_ for α-crystallin (**A**) and HMWA (**B**) on the urea concentration. Black circles—denaturation with the increase in urea concentration, open circles—renaturation with the decrease in urea concentration.

**Figure 2 ijms-24-13473-f002:**
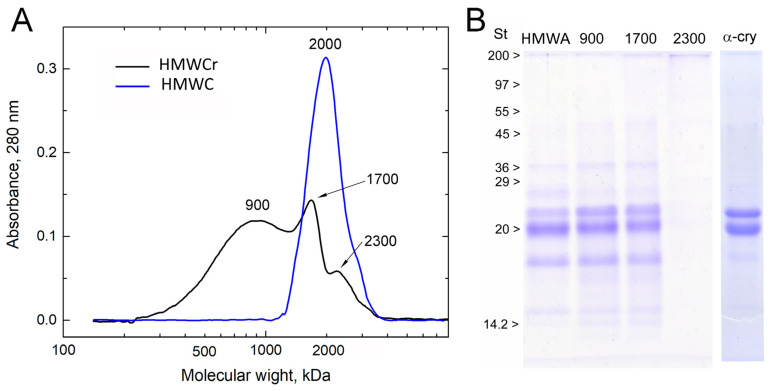
Molecular and polypeptide composition of HMWA and HMWAr. (**A**)—*D*_h_ exclusion chromatography of HMWA and HMWAr, (Column 2.5 × 90 cm, Sephacryl S300HR). The numbers indicate the molecular mass of the fraction. (**B**)—SDS PAGE of HMWA, HMWAr fractions (900—900 kDa fraction, 1700—1700 kDa fraction, 2300—2300 kDa fraction) and α-crystallin (α-cry), St—protein standards.

**Figure 3 ijms-24-13473-f003:**
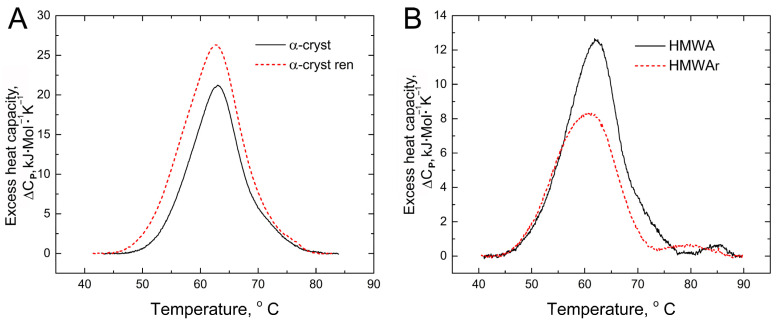
DSC profiles of native and renatured α-crystallin (**A**) and HMWA (**B**). Solid black line—native proteins, and dashed red line—renatured proteins. Protein concentration in all the samples was 1 mg/mL.

**Figure 4 ijms-24-13473-f004:**
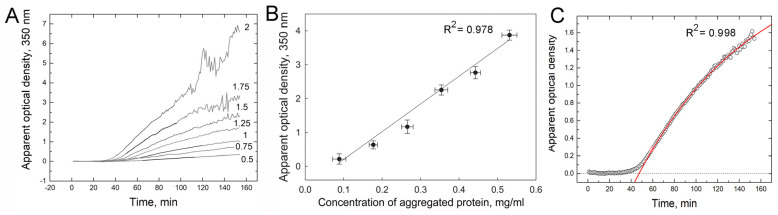
Aggregation of UV-damaged β_L_-crystallin: (**A**)—the dependences of the apparent optical density on time obtained at the varied values of protein concentration. The numbers to the right of the curves show the protein concentration (mg/mL) in the incubation mixture. (**B**)—the dependences of the apparent optical density on the concentration of aggregated protein (black circles—mean values, whisker—SD, the number of experiments *n* = 3). (**C**)—an example of the aggregation curve of UV-damaged β_L_-crystallin (1.5 mg/mL); fitting with the equation of the first-order reaction $A=A_{\lim}\cdot (1-e^{-k_{I}\cdot (t-t_{0})})$.

**Figure 5 ijms-24-13473-f005:**
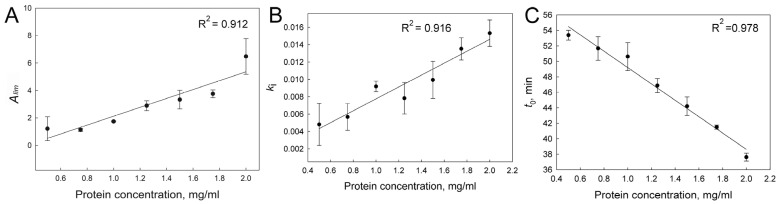
The dependence of main parameters characterizing aggregation of UV-damaged β_L_-crystallin on protein concentration. (**A**)—*A*_max_ dependence; (**B**)—*k*_i_ dependence; (**C**)—*t*_0_ dependence (black circles—mean values, whisker—SD, the number of experiments *n* = 3).

**Figure 6 ijms-24-13473-f006:**
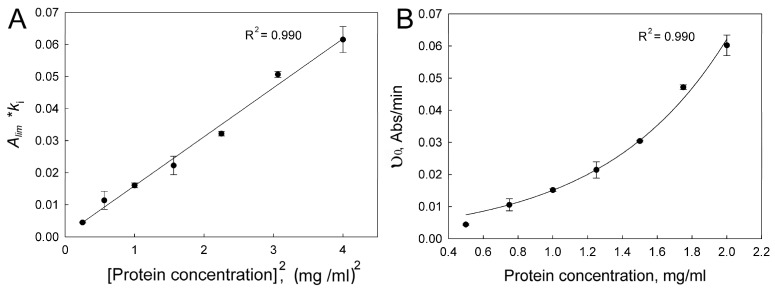
The dependence of initial aggregation rate of UV-damaged β_L_-crystallin on protein concentration. Black circles—mean values, whisker—SD, the number of experiments *n* = 3. (**A**)—fitting with linear function using following coordinates [*k*_I_ ·*A*_lim_]—[P]_0_^2^; (**B**)—fitting with quadratic function υ_0_ = a·[P]_0_^2^.

**Figure 7 ijms-24-13473-f007:**
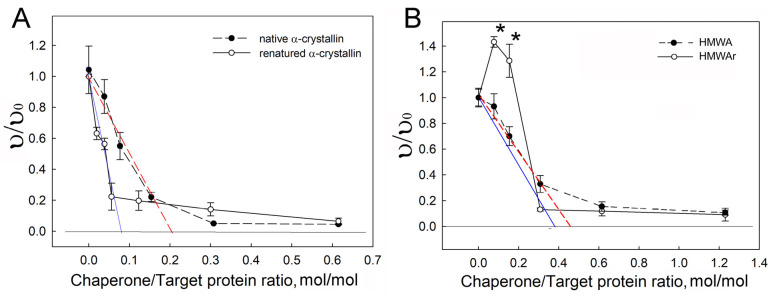
Measure of initial adsorption capacity of native and refolded α-crystallin and HMWA complex. (**A**)—native and refolded α-crystallin; (**B**)—native and refolded HMWA. Black circle—native proteins, unfilled circle—refolded proteins (mean ± SD, the number of experiments *n* = 3). Fitting with linear function: solid blue line—refolded proteins, dashed red line—native proteins. *—*p* < 0.05. Mann–Whitney U nonparametric test with Bonferroni correction.

**Figure 8 ijms-24-13473-f008:**
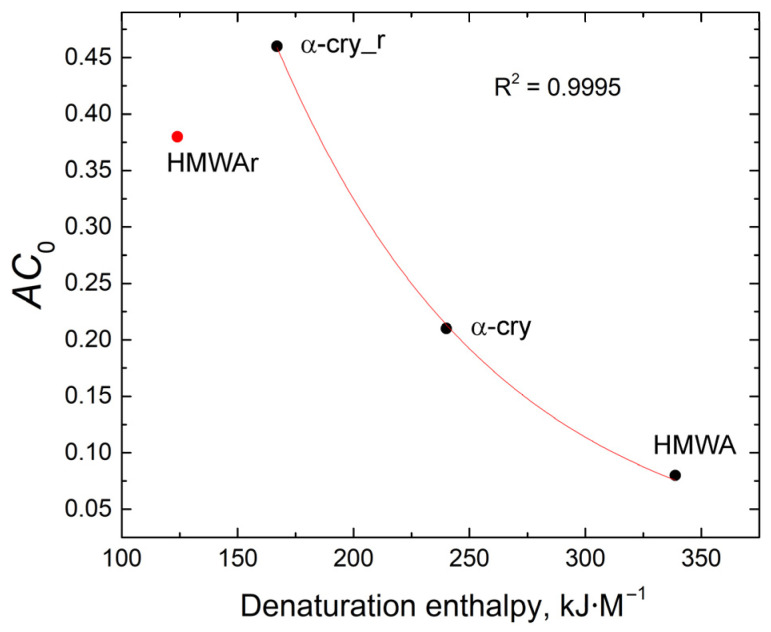
Dependence of the chaperone-like activity on the denaturation energy for the studied proteins.

**Table 1 ijms-24-13473-t001:** Denaturation enthalpy and T_max_ of native and refolded α-crystallin and HMWA.

	α-Crystallin		HMWA	
	Native	Renatured	Native	Renatured
T_max_, °C	63.0	62.7	62.2	60.9
Denaturation enthalpy, kJ∙Mol^−1^	240.1	338.8	167.1	124.2

**Table 2 ijms-24-13473-t002:** Comparison of chaperone-like activity (initial adsorption capacity) of native and refolded α-crystallin and HMWA.

	α-Crystallin		HMWA	
	Abscissa Point of Intersection	Target Protein/Chaperone mol/mol Ratio	Abscissa Point of Intersection	Target Protein/Chaperone mol/mol Ratio
Native	0.21 ± 0.02 ^1,2^	5:1	0.46 ± 0.02 ^1,2^	2:1
Refolded	0.08 ± 0.001 ^3^	13:1	0.38 ± 0.01 ^3^	3:1

^1^—*p* < 0.05, Mann–Whitney nonparametric U test. Comparison of native and refolded proteins. ^2^—*p* < 0.05, Mann–Whitney nonparametric U test. Comparison of native a-crystallin and native HMWA. ^3^—*p* < 0.05, Mann–Whitney nonparametric U test. Comparison of refolded a-crystallin and refolded HMWA.

## Data Availability

Not applicable.

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
