# Peer review of "Refolding Increases the Chaperone-like Activity of αH-Crystallin and Reduces Its Hydrodynamic Diameter to That of α-Crystallin"

_ijms, 2023, doi:10.3390/ijms241713473_

Round 1
Reviewer 1 Report

English is fine but there are a number of typographical errors. However, the logic flow of the manuscript MUST be improved.
Author Response
Reviewers stated that the major issue is the amount of typos and some unclear logical statements, whereas the language of the manuscript is grammatically correct. Therefore, we have performed the thorough spell-checking of the manuscript and significantly rewrote the text to make it more clear and consistent.
Corrections to the text of the manuscript are marked in yellow.
Remark: Data are presented in an unclear manner and it is impossible to understand something. For instance, the title of the manuscript states “refolding restores the initial size and increases a chaperone-like activity of the high molecular weight aggregate of alpha-crystalline”, but Figure 2B unequivocally indicates that refolding does not lead to the same diameter of the HMWA particles (note that there is no explanation in the manuscript of the manner in which renaturation is actually performed). Similarly, the analysis of data in Figure 7B is questionable.
Answer: The title of the manuscript has been corrected. Changes have been made in the logic of presentation.
Remark: The function reported on the y-axis of Figure 3 is wrong.
Answer: We have corrected this annoying error in the names of the axes in Fig. 3.
Remark: Both protein samples consist of different protein aggregates and so the real meaning of DSC profiles is unclear.
Answer: Aggregation does not affect the enthalpy of denaturation because it occurs in time when denaturation is completed. Perhaps in this sentence the reviewer indicates that the native molecule consists of several subunits. Oligomerization also does not affect the enthalpy of denaturation, since it does not change the secondary and tertiary structure of the subunits. For DSC studies we used only solutions of individual proteins, not mixtures of them. So it is unclear what caused this claim of the reviewer.
Remark: The denaturation enthalpy values obtained from the DSC profiles of HMWA are very very small and suggest that something is not perfect![
Answer: α-crystallin really has very low enthalpy of denaturation as it was reported earlier (V.A. Borzova et al. / International Journal of Biological Macromolecules 2015, 73, 84 http://dx.doi.org/10.1016/j.ijbiomac.2014.10.060).
Remark: Figure 4 consists of 3 panels, but the latter have been constructed in a strange manner: panel B should be the consequence of panel A, but the values of protein concentration on the x-axis do not correspond; panel C shows the result of a fitting of one of the curves in panel A, but it is not indicated which one.
Answer: Figure 4B demonstrates the result of special experiment and does not relate to Fig 4A. Please see 3.9. section. Additional clarifications added to the text.
Reviewer 2 Report
I have read the manuscript by Muranov and colleagues and my thinking is that it is unsuitable to be published in its present form. Data are presented in an unclear manner and it is impossible to understand something. For instance, the title of the manuscript states “refolding restores the initial size and increases a chaperone-like activity of the high molecular weight aggregate of alpha-crystalline”, but Figure 2B unequivocally indicates that refolding does not lead to the same diameter of the HMWA particles (note that there is no explanation in the manuscript of the manner in which renaturation is actually performed). Similarly, the analysis of data in Figure 7B is questionable.
Muranov and colleagues assign a great significance to the DSC measurements shown in Figure 3A for alpha crystallin, and Figure 3B for HMWA (note that the function reported on the y-axis is wrong because the output of DSC instruments is the heat capacity not the enthalpy). However, both protein samples consist of different protein aggregates and so the real meaning of DSC profiles is unclear. The suggestion that in the case of alpha crystallin renaturation causes an increase in the fraction of folded protein is a simple speculation unless an experimental evidence coming from a different probe is provided. In addition, the denaturation enthalpy values obtained from the DSC profiles of HMWA are very very small and suggest that something is not perfect!
Figure 4 consists of 3 panels, but the latter have been constructed in a strange manner: panel B should be the consequence of panel A, but the values of protein concentration on the x-axis do not correspond; panel C shows the result of a fitting of one of the curves in panel A, but it is not indicated which one.
The authors have to perform a major-major revision of the manuscript (and of the Figures) to render it suitable to be published.
English is not bad, but has to be improved.
Author Response
Reviewer 2
Corrections to the text of the manuscript are marked in green.
Major issues:
- …”. There are many, many typographical errors”
Answer – Done
- Overall the English is in of itself generally correct but the section need thesis and topic sentences as the structure and logic of the arguments presented are very difficult to follow…
Answer – Done. Significant changes have been made to the “Introduction”, the necessary explanations have been added to the “Results” and “Discussion” section.
3) Remark 1. The authors show in figure 2B that HMWA fully denatured on SDS-PAGE is a-crystalline (also see point 4). However, in figure 3 and table 1, there are different melting temperatures for a-crystaline and HMWA / HMWAr. There does not appear to be a notable unfolding / loss of quaternary structure in any of the data shown so it would appear that unfolding of the tertiary structure of a-crystaline monomers occurs simultaneously with the loss of the larger structure.
Answer: The melting of the tertiary structure of polypeptide molecules often occurs simultaneously with the loss of their quaternary structure. In other cases, dissociation into subunits occurs before the melting of the tertiary and secondary structures. It is well known, that the dissociation into subunits never affects the shape of the DSC profile.
Remark 2. However, for the renatured HMWA, this is lower than for alpha-crystallin by itself. This SHOULD NOT be possible unless there is a notable flaw in the measurements or experimental design. The authors need to address this issue, merely stating that some of the protein is more thermolabile is not adequate. Additionally, the change in size of HMWAr is also concerning and confounds the authors conclusions that it is in a thermolabile state if it matches a-crystaline in size.
Answer 2. Indeed, in the presence of detergents (SDS, urea), both a-crystallin and HMWA dissociate to monomers. However, the monomers that make up HMWA have a greater number of post-translational modifications compared to α-crystallin [Liang JJ, Akhtar NJ. Biochem Biophys Res Commun. 2000, 275, 354, doi: 10.1006/bbrc.2000.3306. Plater ML, et al., Ophthalmic Res. 1997, 29, 421, doi: 10.1159/000268043]. Posttranslational modifications can alter intramolecular bonds that stabilize the tertiary structure of the molecule. For example, oxidation of the SH-group of cysteine is able to shift the DSC peak of tropomyosin by several degrees. Changes in the thermal stability and enthalpy of denaturation of crystallin are evidence of the presence of significant posttranslational modifications in HMWA, that also decrease its chaperon-like activity.
The necessary explanations are included in the text of the manuscript.
4) Remark 1. Hereafter, we will use this term to emphasize that αH-crystallin is a protein particle stochastically assembled from α-crystallin oligomers”. This is a big inconsistency.
Answer Necessary changes have been made to the text.
It should be clarified that in this case a-crystallin is a modified protein. Not only its monomers have post-translational modifications, but the oligomer itself can include damaged lens proteins, for example, g-crystallin.
4) Remark 2. However, as 80% or more of the total protein corresponds to the alpha-crystallin bands, its likely these are just PAGE artifacts or some kind of SDS resistant aggregate (perhaps buried disulfide linkages?) However, they do not demonstrate that these are different proteins which can be done easily by excising the bands followed by proteolytic sequencing. This or another method is required to argue that there are non-crystallin proteins present in HMWA. The authors' statement that „Identification of group 2 and 3 poly-peptides was not carried out as it was not necessary to achieve the main objective of the study.” is not acceptable given this discrepancy.
Answer The formation of covalent intermolecular crosslinks in alpha crystallin, the number of which increases with age, is a well known fact. Some of them are not restored in the presence of DTT, as they have a different nature. In the absence of other proteins, a-crystallin forms dimers, trimers, etc. with oxidative damage [Borzova et al., Int.J.Biol.Macromol., 2015, 73, 84, doi: 10.1016/j.ijbiomac.2014.10.060]. Therefore, the presence of proteins with a mass not a multiple of 20 kDa indicates that other proteins are also included in HMWA. At the same time, taking into account that the main result of the study is the discovery of a relationship between the chaperone-like activity of the protein and the ratio of the denatured and native component in it, the identification of impurity proteins is not necessary.
The necessary explanations are included in the text of the manuscript.
Minor issues:
1) Is 2.2 fold measurable or within the experimental error? This is a small amount and a more
explicit description of why this is significant and clearly measurable should be included.
Answer. Significance levels proving the significance of differences for AC0 of the studied samples are given in the Table 2.
2) The abstract uses the term ”2.2-fold lower” which should be avoided in English due to
vagueness. Give the amount as a % of the standard / control measurements instead. Likewise, with the term “1.2-fold increase” as this can be either 120% or 220% of the baseline amount.
Answer. Done.
3) The sentence “The polypeptide composition of a-crystallin showed no difference from
standard preparations [53]. It consists of aA-crystallin (30%), aB-crystallin (60%) and
truncated -crystallin (10%).” is confusing as the authors use the term a-crystallin twice to
mean both a thing itself and a component of the thing. This makes it difficult to follow the
rest of the paper and a better terminology is required.
Answer. Done. It consists of aA-crystallin (30%), aB-crystallin (60%) and truncated a-crystallin, which is a mixture of aA-crystallin (30%) and aB-crystallin fragments (10%).
4) The authors state “The strong noise at the terminal stage of aggregation at a protein
concentration above 1.25 mg/ml is caused by the fact that the measuring beam of light
passes vertically” is simply incorrect. The issue is most assuredly that absorbtion is the
opposite of transmission and at high optical densities little to no signal from the light beam
is reaching the detector. See for example https://en.wikipedia.org/wiki/Beer
%E2%80%93Lambert_law
Answer.
The noise of the aggregation curve is not stochastic (Fig. 4A), as is typical for electronic devices. The author's link to the Wikipedia article deals with just such noises. Moreover, the SPECTROstar Nano optical density range: 0 to 4 OD and photometric resolution: 0.001 OD. Hence the results were obtained in the linear measurement area. The nature of the noise is described in more detail in the revised version of the manuscript.
“The strong noise at the terminal stage of aggregation at a protein concentration above 1.25 mg/ml is caused by the following circumstances: (i) the precipitation of protein aggregates; (ii) the vertical direction of the measuring beam of light, the fluctuations of which are enhanced upon mixing the sample during the movement of the microplate carrier. It is necessary to note that in further analysis we used the initial parts of the kinetic curves and bL-crystallin concentration 1.5 mg/mL”.
5) Edit the sentence: „Refolding causes a tow-fold decrease in the size of HMWA”
Answer. Done.
6) The methods state that the crystalines are obtained from steers, but later refers to these
animals as bulls which is inconsistent.
Answer. Done.
7) The authors state that: „Refolding reduces the hydrodynamic diameter of the a-crystallin
oligomer from 19.5 to 17 nm. This indicates an increase in the amount of oligomers in the
sample, since they are formed from the same amount of monomers.” however, this does not
follow as compaction and changes in shape may also yield similar results.
Answer. We discussed reviewer’s supposition as follows.
“The decrease in α-crystallin size can be explained by an increase in the compactness of the oligomer. However, an increase in the chaperone-like activity of a-crystallin, on the contrary, requires loosening of the structure of protein chains, i.e. increasing the availability of hydrophobic sites for target protein binding [53].”
Round 2
Reviewer 1 Report
The manuscript is now significantly improved in terms of readability. There are sill a few minor points.
1) Oligomerization, especially for large oligomers or aggregates which would seem to be appropriate here. However, the author's edits to the manuscript largely ameliorate previous concerns.
2) "Urea induced refolding slightly reduces the size of the α-crystallin" is vague to the point of being incorrect. (p.14) The authors should use a specific term (e.g. hydrodynamic radius) instead (as well in the title).
3) p. 5 "α-crystallin andHMWA upon refoldin." should be "refolding"
4) p. 7 " indicates the mechanism of nucleation-dependent" should be "a mechanism"
5) All results must be reported in the past tense. This is because one is reporting what happened previously (when the experiment was performed). example: "The single-hit model of UV-induced denaturation is not performed for these proteins". should be "was not performed". There are numerous examples in the results which must be corrected.
6) p. 10 " its structure is violated during the renaturation process" should read "was changed during" or something similar.
7) The authors still need to fix the spacing issue with hyphenated names such as a-crystallin.
Author Response
First of all, we thank the referee for a thorough and unbiased analysis of our manuscript. Constructive comments and recommendations allowed us to look at our study from a different angle. We took into account all the comments and made the necessary changes to the text, the main of which are marked in yellow.
Comments and Suggestions for Authors
The manuscript is now significantly improved in terms of readability. There are sill a few minor points.
1) Oligomerization, especially for large oligomers or aggregates which would seem to be appropriate here. However, the author's edits to the manuscript largely ameliorate previous concerns.
Answer – For the best understanding of the oligomeric state of samples studied we additionally clarified the protocol for obtaining denatured and renatured samples in the section 3.3. Methods for studying each type of sample are also indicated there.
2) "Urea induced refolding slightly reduces the size of the О±-crystallin" is vague to the point of being incorrect. (p.14) The authors should use a specific term (e.g. hydrodynamic radius) instead (as well in the title).
Answer - Done
3) p. 5 "a-crystallin andHMWA upon refoldin." should be "refolding"
Answer - Done
4) p. 7 " indicates the mechanism of nucleation-dependent" should be "a mechanism"
Answer - Done
5) All results must be reported in the past tense. This is because one is reporting what happened previously (when the experiment was performed). example: "The single-hit model of UV-induced denaturation is not performed for these proteins". should be "was not performed". There are numerous examples in the results which must be corrected.
Answer - Done
6) p. 10 " its structure is violated during the renaturation process" should read "was changed during" or something similar.
Answer - Done
7) The authors still need to fix the spacing issue with hyphenated names such as a-crystallin.
Answer - Done
Reviewer 2 Report
I have read the revised version of the manuscript, and I do not see any changes that can render such a manuscript suitable for publication.
It seems the idea of the authors is to show the relationship between the chaperone-like activity of alpha-crystallin and its high molecular weight version in both “wild-type form” and “urea-renatured form”. Since these two “proteins” consist of different components and aggregates (this is the real situation in the eye lens), experimental studies should be done with great-great care, controlling everything in detail. In the present manuscript, I am still unable to understand what is the urea-renatured form of alpha-crystallin and its high molecular weight version.
For instance, in the DSC measurements:
a. Is the sample subjected to the first heating scan the one that is subjected to the second heating scan?
b. Is the sample labelled renatured a different sample that has been denatured in 8 M urea and then renatured?
c. How has been performed the latter renaturation? (I did the same question in my previous review; this is a fundamental point because all the novelty of the study would be the characterization of the “urea-renatured forms”).
d. The idea to compare the enthalpy change associated with the temperature-induced denaturation of such “proteins” could be correct, but a non-ambiguous definition of the samples would strictly be necessary.
A final point: on reading the abstract it is very-very difficult to understand what has been done in the study!
English is not bad, but it can be improved.
Author Response
We thank the reviewer for a thorough analysis of the manuscript and hope that the revised version of the manuscript has become more understandable and interesting to the reader.
Remarks
It seems the idea of the authors is to show the relationship between the chaperone-like activity of alpha-crystallin and its high molecular weight version in both “wild-type form” and “urea-renatured form”. Since these two “proteins” consist of different components and aggregates (this is the real situation in the eye lens), experimental studies should be done with great-great care, controlling everything in detail. In the present manuscript, I am still unable to understand what is the urea-renatured form of alpha-crystallin and its high molecular weight version.
Reply. The main idea is to show the relationship between the chaperone-like activity and the portion of native protein in both alpha-crystallin and alphaH-crystallin. It was directly stated in the first variant of revised manuscript. We have emphasized this idea in the current variant (revision 2). Reviewer’s remark concerning any differences “between these two “proteins”” stats widely known information and of course has been taken into account during experiment planning.
a. Is the sample subjected to the first heating scan the one that is subjected to the second heating scan?
Reply. We have specified the used experimental protocol for DSC in section “Materials & Methods”, subsection 3.7.
The irreversibility of the thermal transition of proteins was proved by checking the reproducibility of the calorimetric trace. To do this, the sample of the protein was scanned twice in a row without removing it from the calorimeter.
b. Is the sample labelled renatured a different sample that has been denatured in 8 M urea and then renatured?
Reply. We have specified the used experimental protocol for protein denaturation/renaturation in section “Materials & Methods”, subsection 3.3.
Renatured -crystallin and HMWA for the DSC experiments, study of molecular and polypeptide composition and measurement of chaperone-like activity were prepared as follows. Solution of the corresponding native protein in PBS was mixed with 8 M urea (pH = 7.0) and incubated for 4 h. The final concentration of protein was 8 mg/mL, the final concentration of urea was 6 M. Sample of denatured protein was diluted 1/10 (v/v) with PBS and concentrated using Amicon® Ultra 30K devices on a Beckman J-6 centrifuge. The washing procedure was repeated thrice. Final concentration of the renatured protein was 10mg/mL.
c. How has been performed the latter renaturation? (I did the same question in my previous review; this is a fundamental point because all the novelty of the study would be the characterization of the “urea-renatured forms”).
Reply. We have specified the used experimental protocol for protein denaturation/renaturation in section “Materials & Methods”, subsection 3.3.
d. The idea to compare the enthalpy change associated with the temperature-induced denaturation of such “proteins” could be correct, but a non-ambiguous definition of the samples would strictly be necessary.
Reply. The protocol for obtaining denatured and renatured samples has been described in details in the revised version of section 3.3. Methods for studying each type of sample are also indicated.
A final point: on reading the abstract it is very-very difficult to understand what has been done in the study!
Replay. The abstract has been edited.
Round 3
Reviewer 2 Report
The authors have fixed the points raised in my previous comments. I am satisfied with the performed revision and the manuscript can be accepted for publication.
English is not bad; only minor changes can be performed.